# LkARF7 and LkARF19 overexpression promote adventitious root formation in a heterologous poplar model by positively regulating *LkBBM1*

Gui-Yun Tao[1,4], Yun-Hui Xie[1,4], Wan-Feng Li [1], Kui-Peng Li[2], Chao Sun [1], Hong-Ming Wang[3] & Xiao-Mei Sun[1✉]

Cuttage propagation involves adventitious root formation induced by auxin. In our previous study, *Larix kaempferi BABY BOOM 1* (*LkBBM1*), which is known to regulate adventitious root formation, was affected by auxin. However, the relationship between *LkBBM1* and auxin remains unclear. Auxin response factors (ARFs) are a class of important transcription factors in the auxin signaling pathway and modulate the expression of early auxin-responsive genes by binding to auxin response elements. In the present study, we identified 14 *L. kaempferi ARFs* (*LkARFs*), and found LkARF7 and LkARF19 bound to *LkBBM1* promoter and enhanced its transcription using yeast one-hybrid, ChIP-qPCR, and dual-luciferase assays. In addition, the treatment with naphthalene acetic acid promoted the expression of *LkARF7* and *LkARF19*. We also found that overexpression of these two genes in poplar promoted adventitious root formation. Furthermore, LkARF19 interacted with the DEAD-box ATP-dependent RNA helicase 53-like protein to form a heterodimer to regulate adventitious root formation. Altogether, our results reveal an additional regulatory mechanism underlying the control of adventitious root formation by auxin.

[1] State Key Laboratory of Tree Genetics and Breeding, Key Laboratory of Tree Breeding and Cultivation of State Forestry and Grassland Administration, Research Institute of Forestry, Chinese Academy of Forestry, Beijing 100091, China. [2] Guangxi Forestry Research Institute, Guangxi 530009, China. [3] College of Bioengineering and Biotechnology, Tianshui Normal University, Gansu 741000, China. [4]These authors contributed equally: Gui-Yun Tao, Yun-Hui Xie. ✉email: xmsun@caf.ac.cn

Cutting propagation, which can rapidly form numerous adventitious roots (ARs), is an effective step in the asexual propagation of economically important woody species[1]. However, the capacity to form ARs on stem cuttings varies from species to species, and an insufficient rooting ability or poor quality of a newly established root system may cause economic losses[2]. Therefore, it is important to identify and elucidate the underlying molecular regulatory mechanisms of AR formation to improve the efficiency of vegetative reproduction of woody plants.

AR formation and development is a complex biological process involving cell dedifferentiation, differentiation, root primordia formation, and development. These processes are predominantly affected by endogenous hormones and environmental factors[3–5]. It has been reported that auxin is the major growth-promoting hormone for AR initiation[6,7]. A high Indole-3-acetic acid (IAA) concentration is required for spontaneous AR formation in the *Arabidopsis thaliana* hypocotyls during the early steps[8]. Auxin response factors (ARFs), a class of important transcription factors in the auxin signaling pathway, specifically bind to the auxin response elements (AuxREs, TGTCTC) in the promoters of the downstream auxin response genes[9,10]. A typical ARF protein contains a conserved N-terminal B3-type DNA binding domain and a variable middle region with activator or repressor activity[11]. Some ARF proteins also contain a dimerization domain in the C-terminal region[12]. To date, ARFs have been extensively characterized in numerous plant species. For example, there are 23, 25, 31, and 51 ARFs in *Arabidopsis*, *Oryza sativa*, *Zea mays*, and *Glycine max*, respectively[13–16]. ARFs also have been characterized in woody plants, e.g. there are 39 ARFs in *Populus trichocarpa*[17] and 17 ARFs in *Eucalyptus grandis*[18].

In the past few decades, some *ARF* genes have been found to be involved in AR formation. In *Arabidopsis*, three auxin-induced *Gretchen Hagen3* family genes are regulated by AtARF6, AtARF8, and AtARF17 to promote the activation of the COI1 signaling pathway, affecting AR formation[19–21]. In addition, AtARF7/19 regulates *AtLBD16* and *AtLBD18* to positively control AR and lateral root (LR) formation in *Arabidopsis*[22–24]. In rice, OsARF16 activates *CRL1/ARL1* expression for crown root primordia initiation and development[25,26]. In poplar, overexpression of *PeARF17.1* and *mPeARF17.2* (a peu-miR160a-resistant version of *PeARF17.2*) increased the number and length of ARs[27]. In apple, MdSIZ1-mediated SUMOylation of MdARF8 promotes AR and LR formation[28]. However, the underlying mechanisms and the target genes of ARF7/19 regulating AR development are largely unknown in woody species.

BABY BOOM (BBM) is a member of the APETALA2/ETHYLENE RESPONSE FACTOR (AP2/ERF) family, which works as a major regulator of plant cell totipotency and hence promotes cell proliferation and regeneration during embryogenesis[29–31]. For example, BBM functions in cell differentiation and growth along with PLETHORA1 (PLT1), PLT2, and PLT3 proteins in *Arabidopsis*[32,33]. BBM also regulates the transcription of *LEAFY COTYLEDON1* (*LEC1*) and *LEC2* to mediate somatic embryogenesis in *Arabidopsis* in a dose- and context-dependent manner[31], indicating that BBM acts upstream of the other known embryogenesis transcription factors. Interestingly, *BBM* participates in root primordium formation in an auxin signaling pathway. *MtBBM1* was enriched in the root tips of *Medicago truncatula* and responded strongly to auxin with an extreme increase in root-forming callus[34]. The expression of the *LkBBM1* and *LkBBM2* genes from *L. kaempferi* (larch) was enhanced after naphthalene acetic acid (NAA) treatment during adventitious rooting in larch. Overexpression of *LkBBM1* or *LkBBM2* in poplar increased the number of ARs[35,36]. However, the BBM genetic pathway has not been well characterized[37], and

it is unknown how this protein regulates AR formation. Since ARFs are known to be regulators of auxin response and *BBM1* acts downstream of auxin induction during AR formation, the potential relationship between ARFs and *BBM* needs to be investigated.

DEAD-box RNA Helicases (DDXs or RHs) play important roles in auxin-related plant development. AuxREs are distributed in the promoter regions of various *RHs* in longan and soybean[38,39]. Knockdown of *AtRH7/AtPRH75* caused an auxin-related developmental defect phenotype[40]. In addition, it has been demonstrated that *RHs* play roles in root development. For example, tissue culture of root initiation defective1-1 (rid1-1; an *RH* gene) mutant plants showed that these plants were temperature sensitive for callus, AR, and LR formation from hypocotyl explants in *Arabidopsis*[41]. Compared with Col-0, *Atrh31* mutant seedlings had a shorter primary root length and lower fresh weight, while the *Atrh31* mutant lines complemented with the *AtRH31* gene rescued the *Atrh31* mutant phenotype[42]. Loss-of function of the *RH* gene *AteIF4A-1* reduced the proportion of mitotic cells in the root meristem and disrupted the relationship between cell size and cell cycle progression, thereby reducing *Arabidopsis* LR formation[43]. *BnRH24* is a target of miR158 in *Brassica napus* and overexpression of *BnRH24* in *Arabidopsis* reduced root length[44]. These results suggest that there may be interactions between auxin and *RHs*, which may affect AR development.

In this study, we identified the *ARF* genes in the larch genome and studied the roles of *LkARF7/19* and *LkBBM1* in promoting AR formation. We found that NAA increased the expression of *LkARF7* and *LkARF19*, which in turn promoted AR formation by activating *LkBBM1* transcription and forming heterodimers between LkARF19 and LkRH53. To the best of our knowledge, these findings enrich the understanding of AR formation.

## Results

**Identification of *LkARFs* in larch.** To identify the *LkARF* genes in larch, we performed a BLASTP search against the larch genome database using the protein sequences of the conserved domains of ARFs from *Arabidopsis*, *P. trichocarpa*, *Picea abies*, and *Pseudotsuga menziesii*. Ultimately, a total of 14 potential LkARF protein sequences were identified in the larch genome with high protein sequence similarity to the query sequences (Supplementary Data 1). To evaluate the evolutionary relationships among ARFs, a total of 89 full-length ARF protein sequences from *Arabidopsis*, *P. trichocarpa*, *L. kaempferi*, *P. abies*, and *P. menziesii* were used to generate an unrooted phylogenetic tree (Supplementary Fig. 1). The results showed that all the ARFs were grouped into six classes. Based on the phylogenetic tree, LkARFs showed high sequence homology with those of other gymnosperms.

A phylogenetic tree was also constructed for all the 14 LkARFs using protein sequences (Supplementary Fig. 2a), followed by motif analysis using MEME. Motif 1 was annotated as the B3 domain, motifs 2 and 3 were annotated as the Aux_resp domain, while motifs 5, 6, and 7 were annotated as the Aux/IAA domain (Supplementary Fig. 2b). All of the LkARF proteins contain motifs 1 and 2, whereas LkARF16-1, LkARF16-2, LkARF1-1, LkARF1-2, and LkARF2-2 lack motif 5 and motif 7. Notably, members with close phylogenetic relationships showed similar occurrences of the detected motifs.

**The expressions of *LkARF7*, *LkARF19*, and *LkBBM1* during adventitious rooting.** We measured the relative expression levels of the 14 *LkARF* genes in the ARs, stems, stem tips, and leaves of the hybrid larch (*L. kaempferi* × *L. olgensis*) clone D9 using quantitative real-time PCR (qRT-PCR) (Fig. 1a). We found that *LkARF6-3*,

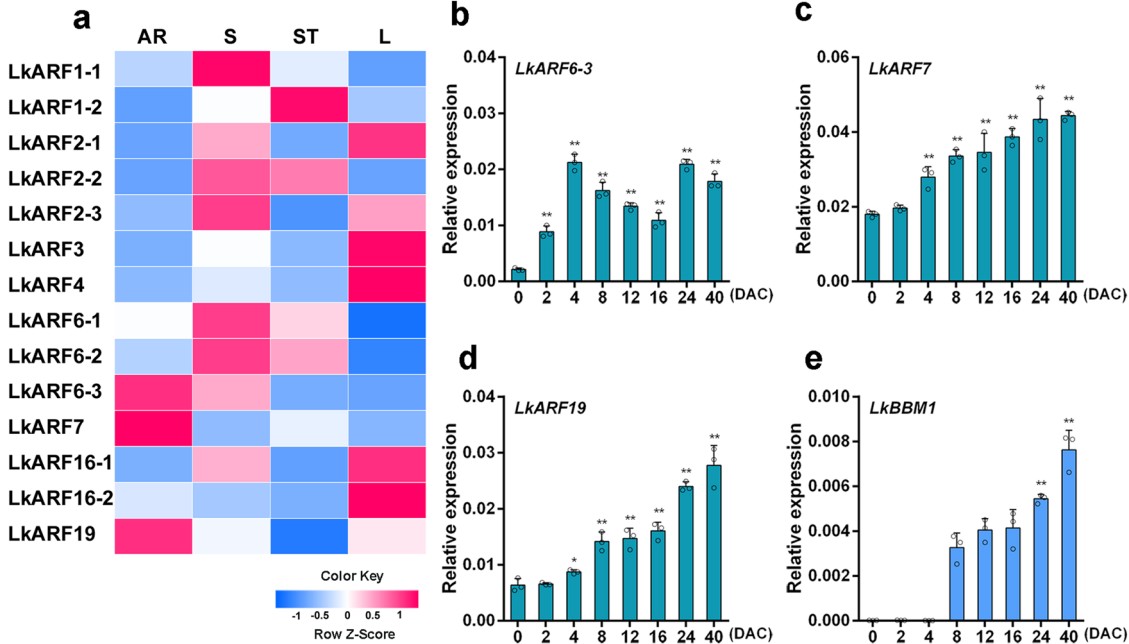

**Fig. 1 Expression analysis of *LkARF* genes in larch. a** Heat map of the expression of 14 *LkARF* genes in different tissues based on qRT-PCR. AR, adventitious roots; S, stems; ST, stem tips; L, leaves. The heat map was made using the Omiscshare tools (http://omicshare.com/tools/) with parameters as default. **b–d** qRT-PCR analysis of *LkARF6-3* (**b**), *LkARF7* (**c**), *LkARF19* (**d**), and *LkBBM1* (**e**) in the basal stem sections (0.5 cm) during in vitro propagation. DAC days after cutting. The *LaEF1A* gene was used as the internal control gene. The error bars represent standard deviations from three biological replicates; *$P < 0.05$ and **$P < 0.01$, $t$ test.

*LkARF7*, and *LkARF19* were strongly expressed in ARs, which might be related to AR formation in larch (Fig. 1a). Then, we analyzed the expression of *LkARF6-3*, *LkARF7*, and *LkARF19* in the D9 clone during adventitious rooting. Even though the relative expression levels of *LkARF6-3* varied from 0 to 40 DAC (days after cutting; Fig. 1b), that of *LkARF7* and *LkARF19* gradually increased following AR formation (Fig. 1c, d). Interestingly, we also found that the relative expression levels of *LkBBM1* gradually increased starting from 8 DAC (Fig. 1e). The expression of *LkARF7* and *LkARF19* were highly correlated with that of *LkBBM1* with Pearson coefficients of 0.94 and 0.97, respectively. Therefore, *LkARF7*, *LkARF19*, and *LkBBM1* were chosen for further analysis.

**The localization of LkARF7 and LkARF19 in the nucleus**. To identify the subcellular localizations of LkARF7 and LkARF19, the 35S::LkARF7-GFP, 35S::LkARF19-GFP, and the control vector (35S::GFP) were transiently transformed into tobacco leaf cells individually. The GFP fluorescence signals were detected exclusively in the nucleus when the LkARF7-GFP and LkARF19-GFP fusion vectors were transformed individually, while that of the control vector were expressed in the nucleus and cytoplasm (Fig. 2). Thus, LkARF7 and LkARF19 are localized in the nucleus.

**The roles of *LkARF7* and *LkARF19* in regulating AR formation**. To investigate the functions of *LkARF7* and *LkARF19* in the promotion of AR formation, we constructed the pBI121-LkARF7 and pBI121-LkARF19 vectors and transformed them into 84 K poplar individually. PCR confirmed that each transgene was present in the genomic DNA of each transgenic line, while qRT-PCR confirmed their overexpression in each transgenic line (Supplementary Fig. 3). We counted the AR numbers in 3-week-old wild-type, the control plants (transformed with empty vector), and the transgenic seedlings (Fig. 3a, b). There was no difference in the AR numbers between the wild-type and control plants. When compared to the control plants, the ARs numbers in *LkARF7*-OE lines (*LkARF7-*

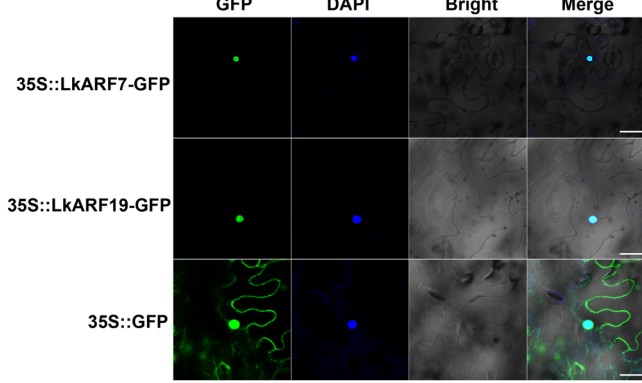

**Fig. 2 Subcellular localization of LkARF7 and LkARF19.** Subcellular localization of LkARF7 and LkARF19 in tobacco leaf cells after tobacco leaf agroinfiltration. Scale bars, 30 μm.

OE3, *LkARF7*-OE4, and *LkARF7*-OE5) and *LkARF19*-OE lines (*LkARF19*-OE4, *LkARF19*-OE14, and *LkARF19*-OE16) increased by at least 46% and 61%, respectively (Fig. 3c). Meanwhile, the total AR length in *LkARF7*-OE lines and *LkARF19*-OE lines increased by at least 26% and 46%, respectively, when compared to the control plants (Fig. 3d). We observed that the ARs in *LkARF7*-OE and *LkARF19*-OE lines appeared one day earlier than that in the control plants (Fig. 3e). In addition, we found that the roots of the 8-week-old *LkARF7*-OE and *LkARF19*-OE lines showed increased fresh (110.0% and 119.1%, respectively) and dry (84.5% and 98.3%, respectively) weight (Fig. 4a–c) when compared to the control plants. The plant height, internode numbers, base internode diameters, and fresh and dry weight of the aerial parts of *LkARF7*-OE and *LkARF19*-OE lines were significantly higher than those of the control plants (Supplementary Fig. 4). Thus, overexpression of *LkARF7* and *LkARF19* significantly promoted AR formation in transgenic poplar.

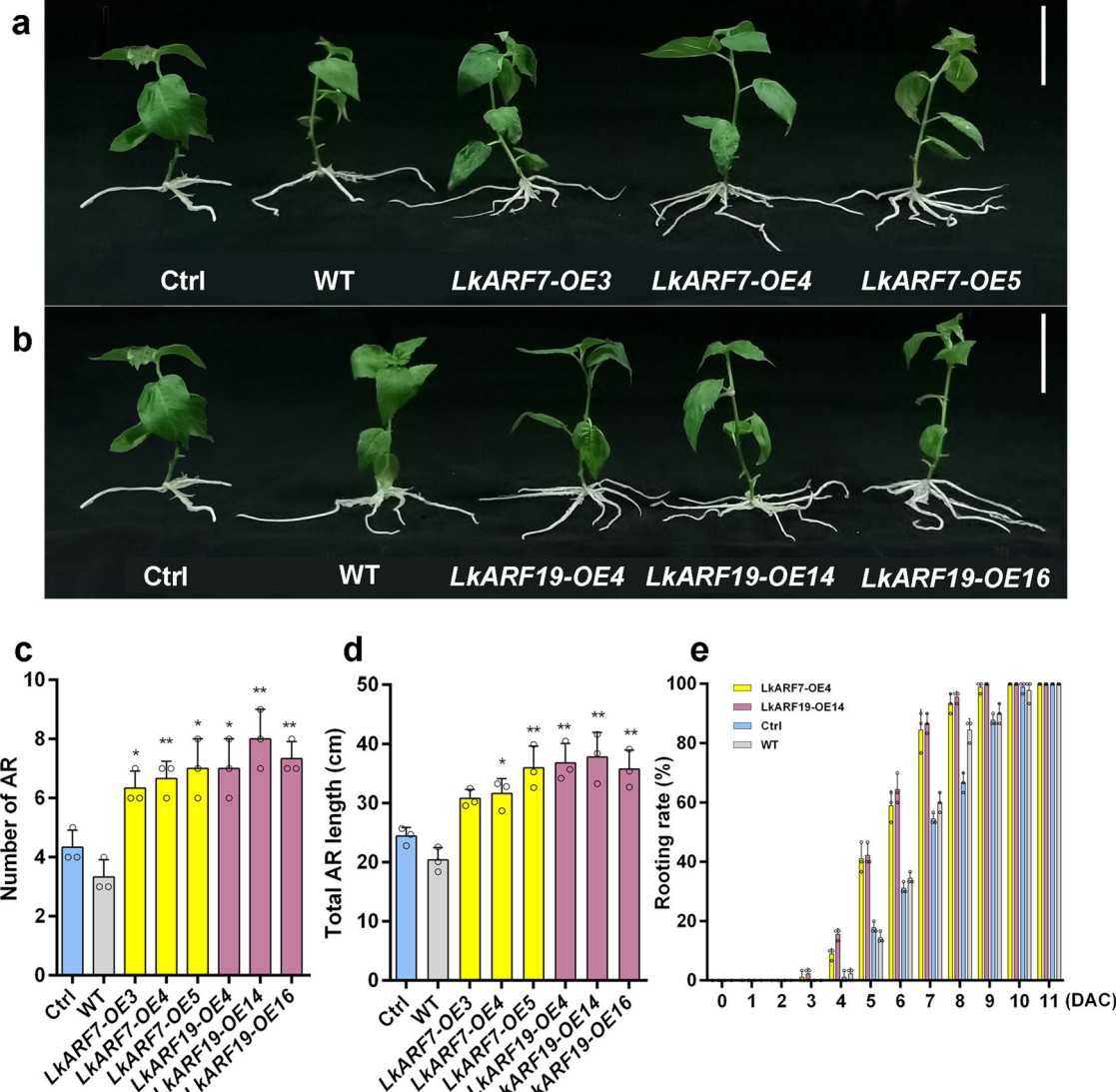

**Fig. 3 Overexpression of *LkARF7* and *LkARF19* promoted AR formation in 84 K poplar. a, b** AR phenotypes on 3-week-old control plants (Ctrl, transformed with empty vector), wild-type (WT), *LkARF7*-OE, and *LkARF19*-OE lines. Scale bars, 5 cm. **c, d** Average AR number (**c**) and total AR length (**d**) in the control plants and overexpression lines of 3-week-old seedlings. **e** Rooting rate of the control plants, WT, and overexpression lines within 11 days following cutting ($n = 30$ per line). DAC, days after cutting. Error bars represent standard deviations from three biological replicates; *$P < 0.05$ and **$P < 0.01$, $t$ test.

**The effect of auxin treatment on *LkARF7* and *LkARF19* expression during AR formation.** To assess whether auxin plays an important role in regulating *LkARF7* and *LkARF19* expression in AR formation, we measured the time-course relative expression levels of *LkARF7* and *LkARF19* in the basal stem sections (0.5 cm in length) of the D9 clone after 0.01 mM NAA treatment. Compared with 0 hours post treatment (hpt), the relative expression levels of *LkARF7* and *LkARF19* at 2, 6, 12, 16, and 24 hpt showed significant increases with an average of 3–8 folds (Supplementary Fig. 5). As a result, auxin treatment significantly increased the relative expression of both genes during AR formation.

**Induction of *LkBBM1* expression by LkARF7 and LkARF19 via promoter binding.** To examine whether LkARF7 or LkARF19 can induce *LkBBM1* expression, the *35S::LkARF7-flag*, *35S::LkARF19-flag*, and the control vector (*35S::flag*) were transiently transformed into the basal stem sections (0.5 cm in length)

of the D9 clone individually. Using qRT-PCR, we found that LkARF7 and LkARF19 induced *LkBBM1* expression by 7 and 10 times, respectively, which were significantly higher than that in the mock treatment (Fig. 5a).

We cloned the 1,727-bp-long promoter fragment of *LkBBM1*. Promoter motif analysis identified several important *cis*-acting elements such as ARR1AT and ABRE (associated with hormonal response), DRE (associated with stress response), and MYBCORE (associated with protein binding) in the *LkBBM1* promoter (Supplementary Fig. 6; Supplementary Data 2). Moreover, three AuxREs (AuxRE-1, AuxRE-2, and AuxRE-3) were identified in the *LkBBM1* promoter (Fig. 5b), suggesting that *ARF* genes may be involved in regulating the transcription of *LkBBM1*.

To investigate whether LkARF7 and LkARF19 can bind to the *LkBBM1* promoter, we carried out yeast one-hybrid (Y1H) assays. We found that 75 mM 3-amino-1,2,4-triazole (3-AT) was the appropriate concentration to inhibit the self-activation (Fig. 5c). The Y187 strain co-expressing pGADT7-LkARF7 and pHIS2-

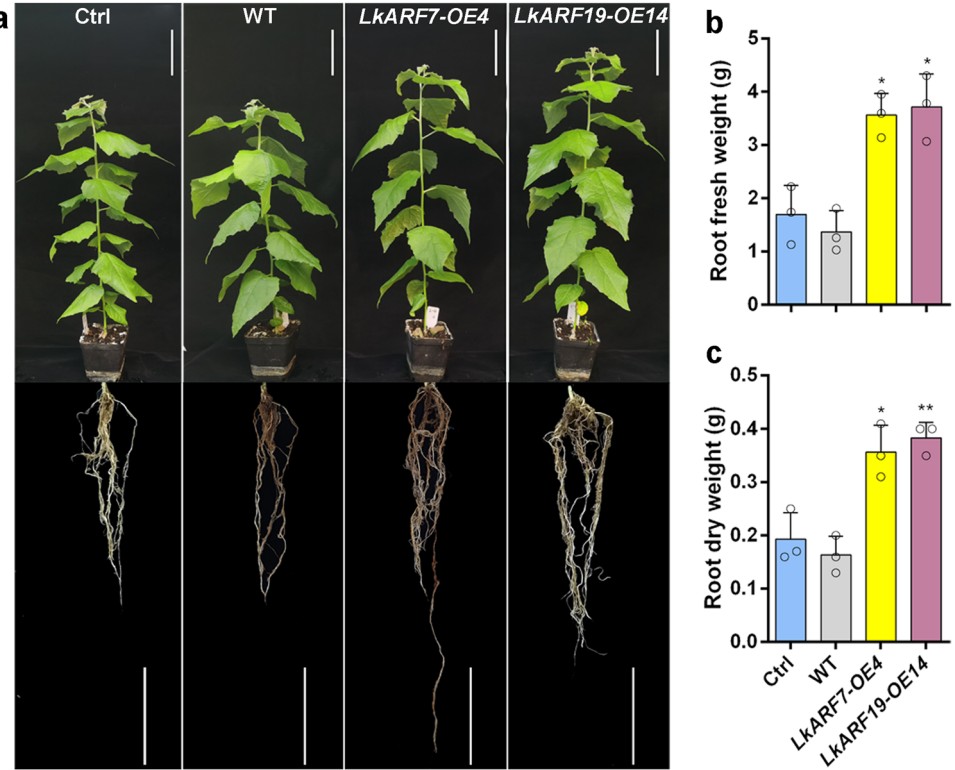

**Fig. 4 Overexpression of *LkARF7* and *LkARF19* promoted root growth in 84 K poplar. a** AR phenotypes on 8-week-old control plants (Ctrl), wild-type (WT), *LkARF7*-OE, and *LkARF19*-OE lines. Scale bars, 10 cm. **b** Fresh and **c** dry weight of roots in the control plants, WT, and overexpression lines. Error bars represent standard deviations from three biological replicates; *$P < 0.05$ and **$P < 0.01$ *t* test.

LkBBM1 and the strain co-expressing pGADT7-LkARF19 and pHIS2-LkBBM1 grew normally on TDO medium containing 75 mM 3-AT, while the negative control strain did not (Fig. 5c). These data indicate that LkARF7 and LkARF19 bind directly to the *LkBBM1* promoter in yeast cells.

To confirm that *LkBBM1* is the direct target gene of LkARF7 and LkARF19, we conducted chromatin immunoprecipitation (ChIP) assays to detect the interactions of LkARF7 and LkARF19 proteins with the *LkBBM1* promoter. Our ChIP-qPCR results showed that the DNA fragments containing each of the three AuxREs in the *LkBBM1* promoter were enriched by overexpressing *LkARF7* and *LkARF19* individually, while the DNA fragment containing the ABRE element was not (Fig. 5d). These results demonstrate that LkARF7 and LkARF19 directly bind to the AuxREs in the *LkBBM1* promoter in vitro.

To further verify the direct interactions between LkARF7/19 and the *LkBBM1* promoter *in planta*, we conducted dual-luciferase assays using tobacco leaf agroinfiltration. When compared to the negative controls, the relative LUC/REN activities driven by the *LkBBM1* promoter reporter were increased by ~2.33 and 2.62-folds when co-transformed with *35S::LkARF7* and *35S::LkARF19*, respectively (Fig. 5e). Furthermore, strong fluorescent signals were detected in the tobacco leaves infiltrated with the *LkBBM1* promoter-driven reporter gene and each of *35S::LkARF7* and *35S::LkARF19* (Fig. 5f). These results indicate that both LkARF7 and LkARF19 directly bind to the *LkBBM1* promoter and activate *LkBBM1* expression in plant cells.

**The interaction of LkARF19 with LkRH53 in the nucleus**. We conducted yeast two-hybrid (Y2H) assays by using the 2-week-old AR's cDNA library constructed in Wang et al.[36] to screen for the LkARF19-interacting proteins. As shown in Fig. 6a, we found

that the AH109 strain containing pGBKT7-LkARF19 and pGADT7-LkRH53 and the positive control strain grew normally on QDO medium (SD/-Trp-Leu-His-Ade) and turned blue in QDO/X medium (SD/-Trp-Leu-His-Ade + X-α-gal), while the negative control strain did not. Thus, LkARF19 directly interacts with LkRH53. To further confirm the interaction between LkARF19 and LkRH53, we conducted the bimolecular fluorescence complementation (BiFC) assays by transforming *LkARF19-cYFP* and *LkRH53-nYFP* into tobacco leaves. The detected YFP fluorescence signals in the nucleus demonstrated that LkARF19 directly interacted with LkRH53 in the nucleus (Fig. 6b).

**The role of *LkRH53* in regulating AR formation**. We used qRT-PCR to measure the relative expression levels of *LkRH53* during cutting propagation of the D9 clone (Fig. 6c). Compared with that at 0 DAC, the relative expression levels of *LkRH53* increased in the basal stem sections (0.5 cm) during cutting propagation by 1.84–2.86 times at 2–40 DAC with the highest expression level at 40 DAC.

We also stably transformed the pBI121-LkRH53 vector into 84 K poplar (Fig. 6d). We used PCR to confirm that the transgene was present in the genomic DNA of each transgenic line and qRT-PCR to confirm its overexpression in the transgenic lines (Supplementary Fig. 3c, f). We found that the average AR numbers and total AR length in 2-week-old transgenic seedlings were significantly higher than the control plants (Fig. 6e). Thus, overexpression of *LkRH53* promoted AR formation in transgenic poplar.

## Discussion

Plant-specific ARF transcription factors regulate plant growth and development such as embryo morphogenesis, vascular tissue

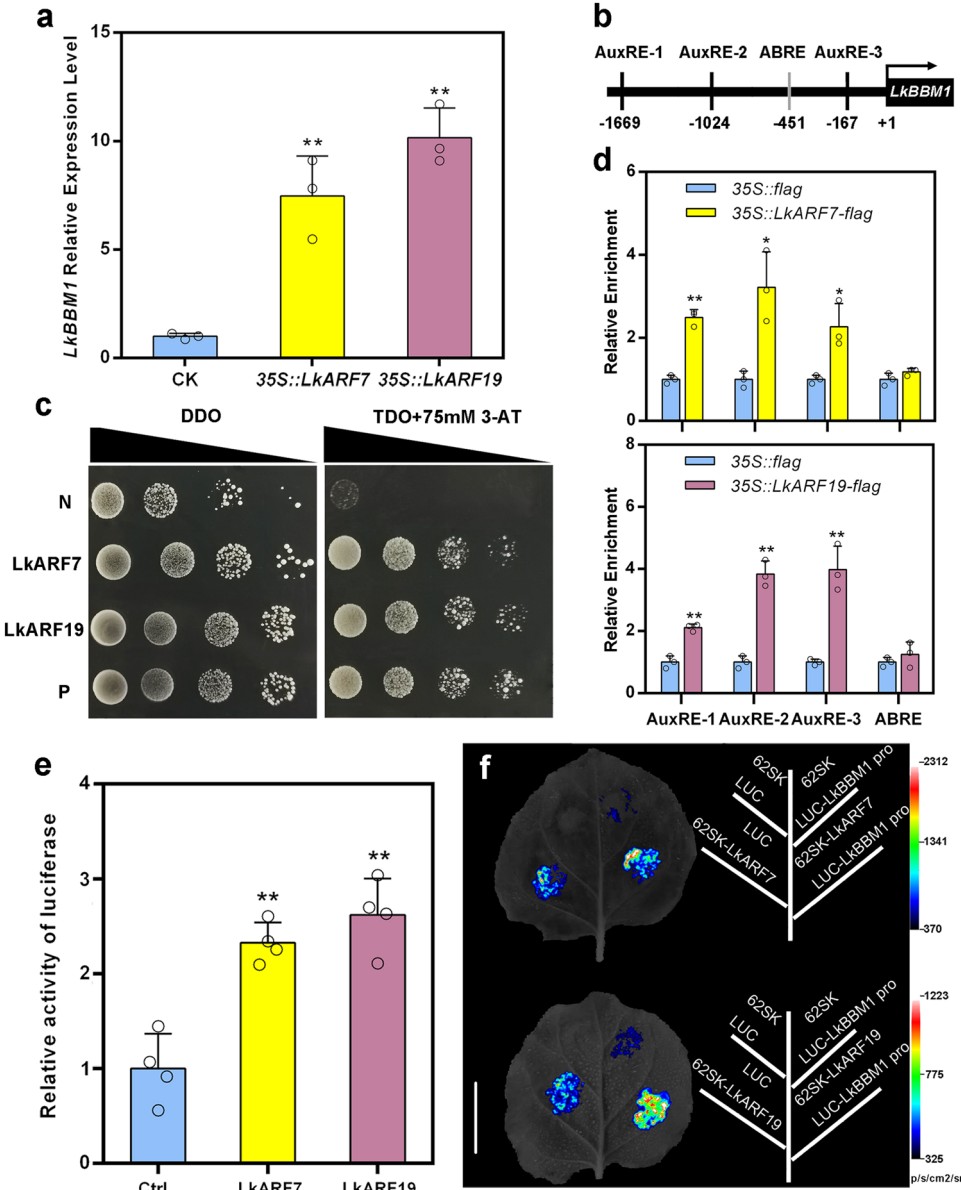

**Fig. 5 LkARF7 and LkARF19 directly bind to the *LkBBM1* promoter and activate its expression. a** Activation of *LkBBM1* expression by transient overexpression of *LkARF7* and *LkARF19*, respectively. The value from the empty vector (CK) was set to 1.0. **b** Yeast one-hybrid analysis of interactions between LkARF7/19 and the *LkBBM1* promoter. P, Positive control (p53HIS2 + pGAD53T); N, Negative control (pHIS2-LkBBM1 promoter + pGADT7); LkARF7 (pHIS2-LkBBM1 promoter + pGADT7-LkARF7); LkARF19 (pHIS2-LkBBM1 promoter + pGADT7-LkARF19). DDO, SD/-Trp-Leu; TDO, SD/-Trp-Leu-His. **c** Diagram of the *LkBBM1* promoter showing the relative positions of three auxin response elements (AuxRE) and one abscisic acid response element (ABRE). **d** Binding of LkARF7 and LkARF19 with the *LkBBM1* promoter using ChIP-qPCR assay. The relative abundance of *LkBBM1* promoter fragments in chromatin was determined by isolation from the *35S::LkARF7-flag*, *35S::LkARF19-flag*, and *35S::flag* lines. **e** Activation of the LUC expression driven by the *LkBBM1* promoter reporter when co-transformed with *35S::LkARF7* and *35S::LkARF19*, respectively. 62SK + Luc-*LkBBM1* promoter was used as the negative control (Ctrl), and its value was set to 1. **f** Representative images of the LUC expression driven by the *LkBBM1* promoter reporter when co-transformed with *35S::LkARF7* and *35S::LkARF19*, respectively. Red and white signals indicate strong binding capacity between effectors and reporters. Scale bars, 4 cm. Error bars represent standard deviations from four biological replicates; *$P < 0.05$ and **$P < 0.01$, t test.

formation, cotyledon development, fiber cell initiation, fruit development, leaf senescence, and floral organ abscission[45–48]. In addition, *AtARF7/19* play roles in leaf expansion, AR and LR formation[49,50]. *SmARF20*, which is homologous to *AtARF19*, regulates LR formation in *Salvia miltiorrhiza*[51]. However, the underlying mechanism of ARF7/19 regulating AR development remains largely unknown. In the present study, we identified 14 *LkARF* genes in larch (Supplementary Fig. 1) and found the expression *LkARF7* and *LkARF19* was induced by auxin during the early stage of AR formation (Supplementary Fig. 5), while

overexpression of *LkARF7* and *LkARF19* promoted AR formation by directly activating the expression of their target gene *LkBBM1* (Figs. 3–5). We also found that LkARF19 could interact with LkRH53, and overexpression of *LkRH53* in transgenic poplar-induced AR formation (Fig. 6e). All of these suggest an additional mechanism regulating AR formation via auxin signaling, LkARF7/19 transcriptionally activation of *LkBBM1*, and interaction between LkARF19 and LkRH53.

In *Arabidopsis*, the loss-of-function mutant of *AtARF7* or *AtARF19* displays altered leaf expansion, AR and LR formation,

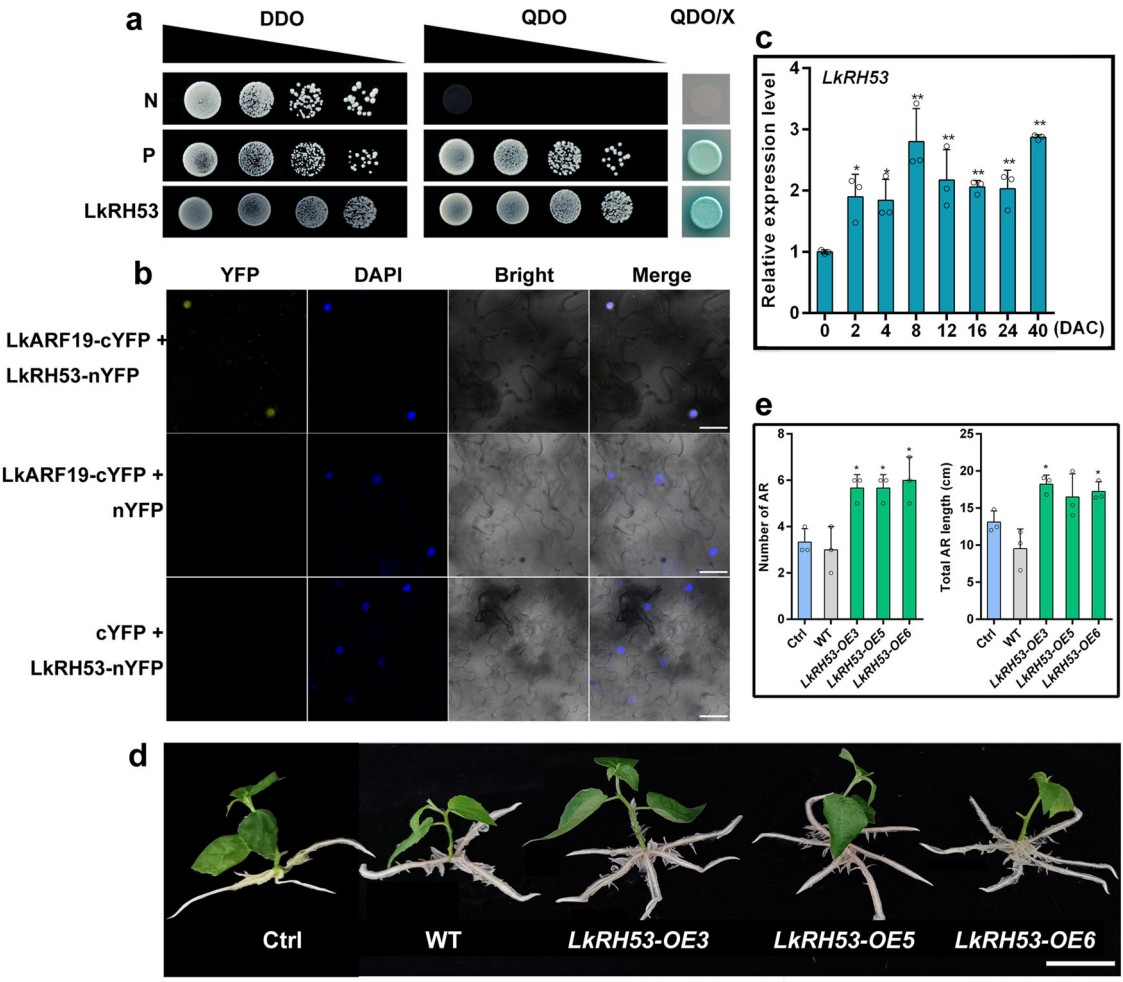

**Fig. 6 Interaction between LkARF19 and LkRH53. a** Yeast two-hybrid analysis of interaction between LkARF19 and LkRH53. N, negative control (pGBKT7-LamC + pGADT7-LargeT); P, positive control (pGBKT7-p53 + pGADT7-LargeT); LkRH53 (pGBKT7-LkARF19 + pGADT7-LkRH53). DDO, SD/-Trp-Leu; QDO, SD/-Trp-Leu-His-Ade; QDO/X, SD/-Trp-Leu-His-Ade + X-α-gal. **b** Bimolecular fluorescence complementation assay analysis of interaction between LkARF19 and LkRH53. LkARF19-cYFP + nYFP and cYFP + LkRH53-nYFP were used as negative controls. Scale bars, 30 μm. **c** qRT-PCR analysis of *LkRH53* in the basal stem sections (0.5 cm) during cutting propagation. DAC, days after cutting. The value on DAC 0 was set to 1. **d** Overexpression of *LkRH53* promoted AR formation in 84 K poplar. Scale bars, 3 cm. **e** Average AR number and total AR length of the control plants, WT, and overexpression lines in 2-week-old seedlings. Error bars represent standard deviations from three biological replicates; *$P < 0.05$ and **$P < 0.01$, $t$ test.

and hypocotyl phototropism[52,53]. We found that *LkARF7* and *LkARF19* were strongly expressed in ARs (Fig. 1a) and the relative expression levels of both genes gradually increased following AR formation (Fig. 1c, d). In this study, we chose to overexpress *LkARF7* and *LkARF19* individually in transgenic poplar, a model species for woody plants. We found that overexpression of each gene significantly increased AR formation (Fig. 3). These findings indicate that *LkARF7* and *LkARF19* play important roles in AR formation.

It is essential to conduct the knockdown and overexpression experiment in larch to provide a more comprehensive understanding of *LkARF7/19* function[54–56]. However, it remains challenging to obtain transgenic larch plants, as a rapid, efficient, and stable transgenic system is still lacking. Model plants such as *Arabidopsis* and poplar have been used to study AR formation and the function of genes from other plants due to the well-established genetic transformation system in them[57–60]. Here, it was found that the sequence similarity of ARF7/19 between larch and poplar was greater than that of *Arabidopsis*, so poplar was used to study the function of *LkARF7/19* in AR formation. Now, transformed larch embryonic cells[61,62] have been obtained and

gene editing has been performed in larch protoplasts[63], giving us more opportunities to study the function of *LkARF7/19* in larch.

It was reported that auxin content could affect the expression of *ARFs*[64]. After 0.5 mM NAA treatment, the expression of *PpARF2B*, *PpARF4*, *PpARF7*, and *PpARF10A* increased rapidly during fruit maturation of peach[65]. In *Arabidopsis*, the expression of *ARF7/19* in hypocotyls is affected by the asymmetric distribution of auxin, which affects the geotropism and phototropism of hypocotyls[66,67]. Previous studies also showed that an auxin signal is required in the induction stage of AR and LR in *Arabidopsis*, *Petunia hybrida*, and *E. globulus*[68–70]. In the present work, we found that the expression levels of *LkARF7* and *LkARF19* significantly increased after the treatment of exogenous NAA for 2–24 h (Supplementary Fig. 5). These data suggest that the participation of *LkARF7* and *LkARF19* in AR formation is promoted by auxin.

Recent research on *BBM* started to focus on root development[34]. It was reported that expression of *LkBBM1* was enriched in ARs and overexpression of *LkBBM1* increased the AR number[35,36]. Our results showed that the expression of *LkBBM1* gradually increased in the basal stem sections during in vitro

propagation (Fig. 1e), which was highly correlated with that of *LkARF7* and *LkARF19* (Fig. 1c, d). We also found three AuxREs in the *LkBBM1* promoter (Supplementary Fig. 6; Supplementary Data 2), which LkARF7 and LkARF19 can directly bind, leading to enhanced *LkBBM1* expression (Fig. 5). Overall, these data suggest that LkARF7/19 promote AR formation by directly activating the transcription of *LkBBM1*.

While our data suggest that *LkARF7/19* promote AR formation by activating transcription of LkBBM1, this hypothesis would require further validation by methods like electrophoretic mobility shift assays to evaluate the relationship between LkARF7/19 proteins and the *LkBBM1* promoter. Likewise, while our Y2H and BiFC data suggest an interaction between LkARF19 and a larch DEAD-box RNA helicase, LkRH53, this interaction should be further verified through other GST pull-down or co-immunoprecipitation assays.

Previous studies have reported that DEAD-box RNA helicases are involved in root development and stress responses[42,71]. In *Arabidopsis*, AtRH53 showed strong promoter activities in root tips in 6-, 9-, and 16-day-old seedlings and a wounding-induced expression in detached leaves[72]. In this study, we identified a larch DEAD-box RNA helicase LkRH53 that interacted with LkARF19 by Y2H and BiFC assays. (Fig. 6a; 6b). Furthermore, we found a high expression level of *LkRH53* during AR formation in larch (Fig. 6c) and the overexpression of *LkRH53* in transgenic poplar promoted AR formation (Fig. 6e). These data suggest that wounding stimulates the expression of LkRH53, which may directly interact with LkARF19 to regulate AR formation.

In conclusion, we propose that auxin-induced expression of transcription factors *LkARF7* and *LkARF19* promote AR formation by forming an LkARF19- LkRH53 heterodimer and transcriptionally activating *LkBBM1* (Fig. 7). These findings provide further insights into the molecular mechanisms of AR formation, which could benefit the molecular breeding of larch.

## Methods

**Plant materials and growth conditions**. D9, a hybrid larch clone (*L. kaempferi × L. olgensis*), was grown in a climate-controlled greenhouse with 22 °C, 16-h/light, and 17 °C, 8-h/dark. Semi-lignified tender stems were propagated by in vitro culture as previously described[35]. Specifically, the newly grown stem tips of 6–8 cm were selected from 2-year-old seedlings, and cultured in medium (L9) with 0.01 mM NAA for 2 weeks. Then, the materials were transferred into NAA-free L9 medium for further culture. The basal stem sections (0.5 cm) on 0, 2, 4, 8, 12, 16, 24, and 40 days, and different tissues, roots, leaves, stems, and stem tips from two-month-old seedlings were collected and then immediately frozen in liquid nitrogen for RNA extraction. In addition, 0.01 mM NAA was added to the L9 medium for 0,

2, 6, 12, 16, and 24 h, and the cultured basal stems were collected and prepared for RNA extraction.

84 K, a hybrid poplar (*P. alba × P. glandulosa*) clone was grown at 23–25 °C under a 16-h/8-h light/dark cycle. *Agrobacterium*-mediated poplar transformation was conducted with leaf discs being used as the explants[73]. Specifically, 2–3 leaves were taken from the tips of 4-week-old 84 K poplar, and 4–5 wounds were cut on the leaves. The leaves were immersed in *A. tumefaciens* solution for 10 min, and co-cultured for 3 days at 25 °C in the darkness on differentiation medium (1/2 Murashige and Skoog (MS) + 0.5 mg/L 6-benzylaminopurine + 0.05 mg/L NAA). After 3 days, the leaves were transferred to differentiation medium with 200 mg/L timentin and 50 mg/L kanamycin for culture. After adventitious buds regenerated, they were cutoff and transferred to rooting medium (1/2 MS + 0.02 mg/L NAA + 0.05 mg/L indole-3-butyric acid) to grow roots.

*Nicotiana benthamiana* was sown onto 1/2 MS solid medium, and 1-week-old seedlings were transplanted into the flowerpot filled with peat and perlite, and grown in an artificial climate room with 22 °C and a 16-h/8-h light/dark cycle. Then, 5-week-old seedlings were used for dual-luciferase assays.

**Promoter cloning and bioinformatics analysis of *LkBBM1***. Larch genomic DNA was extracted from leaves using the TaKaRa MiniBEST Plant Genomic DNA Extraction Kit (Takara, Japan). The cis-regulatory element of the *LkBBM1* promoter was analyzed using the PLACE tool (http://www.dna.affrc.go.jp/htdocs/PLACE/).

**Identification of LkARFs in larch**. The hidden Markov model profiles of the ARF family, including the B3 DNA binding domain (DBD, Pfam02362), AUX_RESP domain (MR, Pfam06507), and AUX/IAA domain (PB1, Pfam02309) were applied to identify ARFs from the *L. kaempferi* genome[74]. Furthermore, 23 *Arabidopsis* (https://phytozome-next.jgi.doe.gov/), 36 *P. trichocarpa* (https://phytozome-next.jgi.doe.gov/), 8 *P. abies* (http://planttfdb.gao-lab.org/index.php) and 8 *P. menziesii* (http://planttfdb.gao-lab.org/index.php) ARF proteins were used as the queries to perform a BLASTP search against the protein sequences of the *L. kaempferi* genome with a cutoff *e* value of $1 \times 10^{-5}$. All identified LkARFs were used for a conserved domain search using CDD (https://www.ncbi.nlm.nih.gov/cdd/) and SMART (http://smart.embl-heidelberg.de/) to confirm the presence of both the intact B3 and Auxin_resp domains. The sequence length, molecular weight, and theoretical isoelectric point of LkARF proteins were calculated using the online tool ExPASy (http://web.expasy.org/protparam/). The subcellular localizations were predicted according to the integration of prediction results of Plant-mPLoc (http://www.csbio.sjtu.edu.cn/bioinf/plant-multi/) and CELLO v2.5 (http://cello.life.nctu.edu.tw/).

**Phylogenetic analysis, conserved motif analysis, and multiple sequence alignment**. For phylogenetic reconstruction of the ARFs family, a total of 89 ARF protein sequences from *L. kaempferi*, *Arabidopsis*, *P. trichocarpa*, *P. abies*, and *P. menziesii* were aligned using ClustalW with default options. Phylogenetic trees were constructed by the neighbor-joining method using MEGA 7.0 software with 1,000 bootstrap replicates[75]. The Multiple Em for Motif Elicitation (MEME) v4.10.2 program (http://meme-suite.org/tools/meme) was applied to identify conserved motifs in the candidate ARF protein sequences. The number of motifs was set to ten. Multiple sequence alignment of ARF protein sequences was analyzed using DNAMAN 9.0 (Lynnon Corp., Canada).

**Promoter cloning and bioinformatics analysis of *LkBBM1***. Larch genomic DNA was extracted from leaves using the TaKaRa MiniBEST Plant Genomic DNA Extraction Kit (Takara, Japan). The cis-regulatory element of the *LkBBM1* promoter was analyzed using the PLACE tool (http://www.dna.affrc.go.jp/htdocs/PLACE/).

**Isolation of RNA and qRT-PCR analysis**. Total RNA was extracted from different tissues using RNAprep Pure Plant Plus Kit (TianGen, China). Total cDNA was synthesized using PrimeScript™ RT reagent Kit (TaKaRa, Japan). qRT-PCR was applied using SYBR Premix Ex Taq II (TaKaRa, Japan) with LightCycler 480 system (Roche, Switzerland). Gene-specific primers for qRT-PCR analysis were designed using Primer3 software (http://bioinfo.ut.ee/primer3-0.4.0/primer3/input.htm) and listed in Supplementary Data 3. LaEF1A1[76] and PtoActin[77] were used as the internal control genes. Three biological replicates were performed in each sample and relative gene expression levels were analyzed using the $2^{-\triangle \triangle Ct}$ method[78,79].

**Subcellular localization of LkARF7 and LkARF19**. The coding sequences (CDS) of *LkARF7* and *LkARF19* (without the stop codons) were amplified and sequenced using the primers listed in Supplementary Data 3. The verified fragments were inserted into the 35S::GFP vector (modified from the PHB vector). The recombinant LkARF7-GFP and LkARF19-GFP fusion and the 35S::GFP control vector were transformed individually into *A. tumefaciens* (GV3101), which were used in the leaves of 5-week-old tobacco seedlings for tobacco leaf agroinfiltration. GFP fluorescence signals in the leaves were visualized under a confocal laser scanning

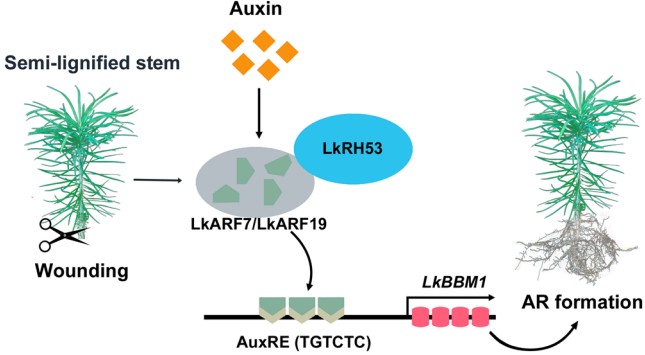

**Fig. 7 A proposed model of the role of LkARF7 and LkARF19 in AR formation.** The expression of *LkARF7* and *LkARF19* increases after the semi-lignified stem is cutoff. Auxin also promotes the expression of *LkARF7* and *LkARF19*. LkARF7 and LkARF19 bind to AuxREs and promote *LkBBM1* transcription to regulate AR formation. In addition, LkARF19 interacts with LkRH53 to form heterodimers to regulate AR formation.

microscope (LSM710, Karl Zeiss, Germany) after 48 h of infection. All transient expression assays were repeated at least three times.

**Transient transformation assay of larch**. The full-length CDS of *LkARF7* and *LkARF19* were cloned into the pBI121 vector containing the flag reporter gene to obtain the fusion constructs *35S::LkARF7-flag* and *35S::LkARF19-flag*. The resulting vectors were transformed into the *A. tumefaciens* (GV3101) individually and then used for transient transformation of the freshly-grown seedlings of D9 clone[80]. Transient transgenic seedlings were cultured on 1/2 MS solid medium under 22 °C, 16-h/light and 17 °C, 8-h/dark for 48 h and then collected for detecting the expression of *LkARF7*, *LkARF19*, and *LkBBM1*. The primers are listed in Supplementary Data 3. Three biological replicates were used for each vector.

**Y1H assays**. The full-length CDS of *LkARF7* and *LkARF19* were cloned into the prey vector pGADT7 individually, and the *LkBBM1* promoter was cloned into the bait vector pHIS2. The combinations of the prey vectors and bait vectors were co-transformed into the Y187 yeast strain, which was grown on SD/-Trp-Leu medium for 3 days at 30 °C and then transferred onto SD/-Trp-Leu-His + 75 mM 3-AT for further incubation at 30 °C. p53HIS2 + pGAD53T were used as the positive control, while pHIS2-LkBBM1 promoter + pGADT7 were used as the negative control.

**ChIP assays**. Transient transgenic larch plants were used for ChIP experiments, which were performed as previously reported[81]. The plants were crosslinked with 1% formaldehyde, pulverized in liquid nitrogen, and then 1% SDS cracking buffer was added for ultrasonication. The sonicated chromatin was immunoprecipitated with anti-flag antibody (1:100 dilution, CWBIO) or without the anti-flag antibody. Each DNA sample was analyzed by ChIP-qPCR to determine the enrichment of promoter fragments. *LaEF1A* was used as the internal control[76]. The primers used are listed in Supplementary Data 3.

**Dual-luciferase assays**. The transactivation roles of LkARF7 and LkARF19 on the target *LkBBM1* promoter were assessed by dual-luciferase assays as previously described[82]. The full-length CDS of *LkARF7* and *LkARF19* were inserted into pGreen II 62SK vector individually, and the promoter fragment of *LkBBM1* was cloned into pGreen II 0800-LUC vector. Each resulting construct was transformed into the *A. tumefaciens* (GV3101-psoup) and transiently expressed in 5-week-old *N. benthamiana* leaves. The Dual-Luciferase® Reporter Assay System (Promega, USA) was used for detection of LUC and REN activities with a dual-luciferase reporter assay system (Promega, GloMax 20/20 Luminometer, Wisconsin, USA). Dual-luciferase assays were performed with at least four biological replicates. The luciferase complementation imaging assay was applied with the Newton 7.0 Bio Plant Imaging System (Vilber Lourmat, France).

**Y2H assays**. An Y2H library from the AR cDNA of the D9 clone was constructed in Wang et al.[36]. The CDS of *LkARF19* was cloned into the bait vector pGBKT7 and transformed into the yeast strain AH109 together with the prey vector pGADT7 to evaluate self-activation. Then protein-protein interactions were screened following the manufacturer's instructions (Clontech, USA). The cDNA library was screened on SD/-Trp-Leu, SD/-Trp-Leu-His, and SD/-Trp-Leu-His-Ade/X-α-Gal medium to identify LkARF19-interacting proteins. The positive clones interacting with LkARF19 were inserted into the prey vector pGADT7 and co-transformed into AH109 with bait vector pGBKT7-LkARF19 to confirm the interactions. The bait vectors pGBKT7-p53 and pGBKT7-LaminC were co-transformed into AH109 with the prey vector pGADT7-LargeT as positive and negative controls, respectively.

**BiFC assays**. The CDS of *LkARF19* (without the stop codon) was cloned into the 35S::SPYCE vector to obtain the fusion construct LkARF19-cYFP. The CDS of *LkRH53* (without the stop codon) was inserted into the 35S::SPYNE vector to obtain the fusion constructs LkRH53-nYFP. The combination of LkARF19-cYFP + LkRH53-nYFP was transformed into the *A. tumefaciens* (GV3101) and transiently expressed in tobacco leaves. The YFP fluorescence was detected by confocal laser scanning microscopy (LSM710; Karl Zeiss, Jena, Germany). The primers used are listed in Supplementary Data 3.

**Overexpression of *LkARF7*, *LkARF19*, and *LkRH53* in 84 K poplar and phenotypic analyses**. The full-length CDS of *LkARF7*, *LkARF19*, and *LkRH53* were amplified and cloned individually into the binary vector pBI121 with the help of *XbaI/SmaI* to generate the *35S::LkARF7*, *35S::LkARF19*, and *35S::LkRH53* vectors. The constructs were then transformed into *A. tumefaciens* (GV3101) and used to transform 84 K poplar[83]. After screening on a selection medium with 50 mg/L kanamycin, transgenic poplar lines were identified through PCR and qRT-PCR. The primers are listed in Supplementary Data 3. Three independent lines were used for each construct for further study. Phenotype analysis of AR number and length in the 3-week-old control (Ctrl, transformed with empty vector), WT, and transgenic poplar were conducted. The plant height, stem diameter, and root fresh and dry weight were measured and the internode numbers were counted in the 8-week-

old plants. Significant differences relative to control plants were determined by Student's *t* test.

**Statistics and reproducibility**. Statistics, which *n* = 3 biologically independent samples were conducted with multiple *t* test using GraphPad Prism v6.02 (GraphPad, America), and asterisks and double asterisks indicate *P* < 0.05 and *P* < 0.01, respectively. The heat map and bar charts were generated using the Omiscshare tools (http://omicshare.com/tools/) and GraphPad Prism v6.02 with the default parameters, respectively. For the dual-luciferase assays, the detection of relative LUC/REN activities was carried out with four biological replicates to ensure the reliability of results. The phenotypic assessment of each poplar overexpression line was independently repeated at least three times. All the qRT-PCR assays were performed with three biological replicates.

**Reporting summary**. Further information on research design is available in the Nature Portfolio Reporting Summary linked to this article.

## Data availability

The sequences of *LkARF7*, *LkARF19*, and *LkRH53* were deposited in NCBI with GenBank accession numbers ON529550, ON529549, and ON529551, respectively. The uncropped gels of *LkARF7*-, *LkARF19*- and *LkRH53*-overexpression transgenic poplar lines are shown in Supplementary Fig. 7. The source data behind the graphs in this study are listed in Supplementary Data 4.

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

## Acknowledgements
This work were supported by the Forestry Industry Research Special Funds for Public Welfare Projects (201504104) and the Special Funds of Research Institute of Forestry (LYSZX202002).

## Author contributions
G.-Y.T., Y-H.X., and X.-M.S. conceived and designed the research. G.-Y.T. performed the experiments and wrote the manuscript. G.-Y.T., Y.-H.X., W.-F.L., K.-P.L., H.-M.W., C.S., and X.-M.S. analyzed the data. W.-F.L. helped write the manuscript. All authors read and approved the final manuscript.

## Competing interests
The authors declare no competing interests.
