## [Peer Review File · Communications Biology]

Reviewers' comments:

Reviewer #1 (Remarks to the Author):

The manuscript by Tao et al. presented that the transcription factors LkARF7 and LkARF19 promote adventitious root formation by positively regulating LkBBM1, which is essential for understanding adventitious root (AR) formation. This study appears to be novel and with good quality, and highlights key findings in the field of gymnosperm research. Before recommending acceptance I must also request authors addressing the following improvements.

- 1) line 43: Please provide the full names of LkRH53 and LkCIP7.
- 2) line 96 to Line 107: State the relationship between ARFs and DEAD-box family genes
- 3) line 99: The "*Physcomitrella patens*" and "*Arabidopsis*" in line 99, the "rh31" in line 103 and line 104, the "*Arabidopsis*" in line 106, the "cop1" in line 116, should be in italics.
- 4) line 102-104: Change the "cold and drought stress" in line 102 to "cold and drought stresses" Transgenic line could be either overexpression line or loss-of-function line depending on the plasmid you transferred. So the "transgenic line" in line 104 should change to "the overexpression line" or " the complemental line"
- 5) line 108 to Line117: Elaborate the relationship between ARFs and COP1-interacting proteins (CIPs) genes.
- 6) line 120: You could not say "LkARF19 interacted with LkRH53 and LkCIP7 to regulate AR formation", because you did not show any evidence that the interaction of LkRH53 and LkCIP7 with LkARF19 is necessary to its AR formation-function.
- 7) line 162: Change the "LkBBM1 began to be expressed at 8 DAC (day after cutting) and then showed an upward trend" to "the expression of LkBBM1 begins at 8 DAC (day after cutting) and gradually increases".
- 8) line 165 to Line 166: There should be a summary sentence explaining why LkARF7 and LkARF19 were studied.
- 9) line 189-191: Change the "The Y187 yeast strain (pHIS2-LkBBM1) was cultured on SD/-Trp-Leu-His (TDO) for screening concentrations of 3-AT and indicated that 75 mM 3-AT was the appropriate concentration" to "The Y187 yeast strain expressing pHIS2-LkBBM1 and pGADT7 empty vector was cultured on SD/-Trp-Leu-His (TDO), and the result showed that that 75 mM 3-AT was the appropriate concentration to inhibit the self-activation". Change the "We found that two Y187 strains, (pGADT7-LkARF7/pHIS2-LkBBM1 and pGADT7-LkARF19/pHIS2-LkBBM1)," to "the Y187 strain co-expressing pGADT7-LkARF7 and pHIS2-LkBBM1, and the strain co-expressing pGADT7-LkARF19 and pHIS2-LkBBM1"
- 10) The subcellular of ARF/BBM and the transcription regulation of BBM1 by ARF should be in separated chapters. In the same time, the fig2A and fig2B-C should in separated figures too. The transcription regulation of BBM1 by ARF could move to the next chapter.
- 11) line 204-212: This assay is named "dual-luciferase assays", NOT "dual-luciferase assays and luciferase complementation imaging assays". CANCEL "in firefly luciferase complementary imaging assays" in line 210 at the same time.
- 12) line 208: Change "a large number of red and white fluorescent signals" to "strong fluorescent signal"
- 13) line 218: Cancel the "In general, the phenotype of LkARF19 OE lines is superior to that of LkARF7 OE lines.", which is not scientific (how you define "superior") ?
- 14) As is shown in fig 4E, the difference of rooting rate between LkARF7-OE, LkARF19-OE lines and Ctrl appeared at 3 DAC, but in line 162 and figure 1E, you mentioned "LkBBM1 began to be expressed at 8 DAC", which is far late behind 3 DAC. These data indicating that it is not the BBM1 that should responsible for the rooting rate different caused by ARF7/19. How do you explain it?
- 15) line 227: The description "The transgenic lines of OE LkARF7 and LkARF19 were larger than Ctrl" is not scientific and precise. How you define LARGER of a seedling? The leaf? Root length? Width of stem? Or the height?
- 16) line 240: Change "Figure 9A" to "Figure 6A"
- 17) line 255: "Because the interaction between LkARF19 and LkRH53 is stronger (than CIP7)". but we could not conclude it from the Y2H data and BIFC data.
- 18) line 259: "LkCIP7 is involved in regulating AR formation." in fact, you do not show sufficient evidence to get this conclusion. The protein interaction data and the expression pattern of CIP7 is not sufficient.
- 19) The paper contains repeated points in the different sections: Results-Methods. Please modify it

appropriately.

20) line 378: At what time were the roots, leaves, stems, and stem tips collected.

21) line 382: What is the method of poplar stable transformation? Describe the method, or provide relevant references.

22) line 438: What are the culture conditions of tobacco?

23) line 455 to Line 457: This sentence is not clearly described. It is suggested to describe the transient expression and ChIP methods separately.

24) In Figure 1, what software is used to make the heat map, what are the parameters?

Reviewer #2 (Remarks to the Author):

This manuscript describes ARF transcription factors as important auxin-induced regulators upstream of BBM in the regulation of adventitious root formation in *Larix kaempferi*, a woody species of economical interest.

The authors state that auxin is important for the formation of adventitious roots and they have earlier identified

BABYBOOM as an important regulator of AR formation in *Larix*. The data in the manuscript add to the model that certain ARFs (7-19) interact directly with BBM to regulate its transcription levels and are hence involved in AR formation. It furthermore places 2 other unknown transcription regulators in the AR pathway.

The experiments are sufficiently well executed and data are correctly represented. A reasoning for transforming poplar and not larch is lacking so the importance of the phenotypes observed in poplar is unclear. The article is well written and does not contain a lot of mistakes. The authors should however, check consequently if genes and mutants are written in italic and proteins not. Throughout the manuscript there are mistakes in this formatting. My major problem is the lack of novelty in the conclusion, since ARFs are already long-known to be involved in adventitious root formation and in their role as direct transcription regulators upon auxin perception it is logic that they would act upon different other transcription factors that regulate cell divisions and identity necessary for AR formation (such as BBM). I have included some suggestions, questions and comments below. Certain comments are essential to be answered for the manuscript to be published. Moreover, I think the authors would benefit from reconsidering the submission of this manuscript to a more specialized plant journal where the visibility will be higher and hence the impact for the scientific community will increase as well.

abstract

I think the termination cutting seedling is confusing.

l38 remove that

l40 expression is not the correct word. The interaction with ARFs promotes transcription of a gene.

l45 ... novel regulatory mechanism underlying the control of AR formation by auxin IN *LARIX KAEMPFERI*.

To me the statement, downstream of auxin signaling is not clear enough - are known auxin receptors studied in *Larix*?

ARFs are known to act downstream of auxin perception but they are a major part of auxin signaling so I would never name this downstream of auxin signaling.

keywords: all genes should be italic (I agree that above where you mention interaction it is okay that they are not italic since there you speak on interacting proteins)

introduction

l56-57 it is unclear what is meant by cell recombination and formation

l57 it is better not to start a sentence with and

l61 auxin response FACTORS (never singular unless a specific one is indicated in the sentence)

l64 if you write receiving here, it is as if ARFs are perceiving the auxin signal. That is not true, they act downstream of auxin receptors (TIR1/AFBs).

175 In addition, IN ARABIDOPSIS THALIANA,

I think the species should be added since it is confusing when model and woody species are mixed. 179 it is not true that the underlying mechanisms are not known. Probably for woody species this statement is more correct, but for models such as Arabidopsis thaliana, different approaches have identified important downstream actors (for example: Lee et al, 2019 BMC Plant Biology; Li 2021, Front Plant Sci; Yin et al, 2020 Plant Phys).

I do think that this statement should be removed or specified. Also, the lack of references to literature in lateral or adventitious root formation and involved ARFs, indicates that the authors might need to do some more background research in the field to well establish their novelty and the importance of their research compared to what is done and known from other species.

181 mention clearly that BBM is a transcription factor, which regulates totipotency in plant cells and hence has a known role in promoting cell proliferation and morphogenesis during embryogenesis 184 mention clearly that BBM is found to act upstream of the other known embryogenesis transcription factors

188 change well to a scientifically more correct description of the response to auxin

193 instead of the conclusion stated:

Since ARFs are known regulators of auxin response and BBM acts downstream of auxin induction during AR formation, we investigate the potential relationship between BBM and ARFs.

196-117 the link to RH and COP1 is not sufficiently explained and not sufficiently smooth or clear. Maybe add something among these lines: In order to investigate potential processes involved in AR formation downstream of BBM, we investigated RH and COP1.

results

1125 ... which genes regulate LkBBM1... (i hope the authors were aiming to identify multiple genes and not just one)

1130 ... suggesting that ARF genes COULD be involved in the regulation OF TRANSCRIPTION of LkBBM1.

1143 you mention members of the gene family, but construct a tree using protein sequences  delete gene from your sentence

1188-195 which controls were used for this Y1H setup?

1204 remove the

1246 for the BIFC assay a negative control ARF would be of interest

1252 how significant is an induction of 1.84x?

1278 the induction of auxin on BBM expression does not indicate a direct downstream effect. To conclude this, one should show that either

- auxin biosynthetic mutants express less BBM

- in an auxin receptor-mutant (tir1/afb) there is no auxin effect on BBM expression

1284 ARFs, plant specific transcription factors, regulate...

1302-304 this is indeed the explanation of auxin signaling and I would mention this much higher up in your manuscript's introduction or when you introduce ARFs as potential expression regulators.

1305 2h is not rapidly

1308 expression IS affected

1309 These DATA suggest that...

1312-313 Previous studies have shown that auxin signaling is required in the induction stages of LR and AR

1313-315 Our data in poplar are consistent with this, since overexpression of LKARF7 and LKARF19 significantly increase the AR numbers.

1320 remove momentous and replace by an important.

Based on this statement, did the authors observe shoot growth phenotypes in the overexpression poplar lines to substantiate this statement?

1329 see earlier, I find this an overstatement since downstream molecular actors are known.

1332-334 expression pattern points more commonly to cellular expression and not to timely expression. Please specify because your data point to a timely relationship in expression of both factors.

1337-343 as you state here, a lot of downstream genes in the BBM signaling are known and loads of them have an important action in root development. Hence the statement of a lack of molecular mechanisms is untrue and should be related to this or mentioned specifically for larch.

1344 it remains unclear why these targets were chosen and not more logic root-developmental

genes as SHR or SCR that were mentioned before.

Fig6 Figure outline is not consistent with the other graphs in the manuscript, they have not received a border (check and adjust everywhere accordingly)
BIFC interaction is not sufficiently clear due to the lack of signal in the YFP channel

Reviewer #3 (Remarks to the Author):

The manuscript entitled "Larch transcription factors LkARF7 and LkARF19 promote adventitious root formation through positive regulation of LkBBM1" by Tao et al. describes the characterization of the auxin response factor (ARF) family of *Larix kaempferi* × *L. olgensis* and identifies LkBBM1 as a target gene of two members of this family. LkBBM1 is a transcription factor of the AP2/ERF family, homologous to the BBM gene of *Arabidopsis* with recognized function in somatic embryogenesis. The participation of LkBBM1 in larch adventitious rooting has been demonstrated in a previous work. In addition, the function of both ARFs in adventitious rooting is studied, using poplar as an experimental system. Their interaction with two other genes is also described. The experimental work is correct, and the methodology is sound. However, certain weaknesses have been detected that should be reviewed before proceeding to publication.

- 1.- DAPI staining is missing in Figure 2A. In addition, the results described in the text corresponding to Figures 2B and 2C are not consistent with those represented in the Figures.
- 2.- It is mandatory to include an additional empty vector control in the poplar gene transformation assays (Figures 4, 5, 6 and S5). Transformations with empty vectors can increase rooting ability in certain species.
- 3.- The increase in biomass, measured as fresh weight or dry weight of the aerial part of the plant and of the root system as a whole, may be a consequence of the increase in the number and length of roots, which could indirectly favor stem growth. No analyses on specific biomass indicators are described that would allow us to conclude on the effect of these genes on biomass beyond the increase in the number and length of roots.
- 4.- The image quality in Figure 6B should be improved. For example, the YFP is difficult to see.
- 5.- Some sections of the discussion seem to have diverged from the main point. Please, streamline the discussion by removing items that are not directly related to the data and consider removing this content from the discussion section that is more appropriate to be in the introduction section.

We sincerely express our deepest gratitude to the editors and three reviewers for their interests in our manuscript and good suggestion, we have modified accordingly.

Reviewer #1

Reviewer's summary-comment:

The manuscript by Tao *et al.* presented that the transcription factors LkARF7 and LkARF19 promote adventitious root formation by positively regulating LkBBM1, which is essential for understanding adventitious root (AR) formation. This study appears to be novel and with good quality, and highlights key findings in the field of gymnosperm research. Before recommending acceptance I must also request authors addressing the following improvements.

Comment 1:

Line 43: Please provide the full names of LkRH53 and LkCIP7.

Response:

We have done. Please see Line 43 and Line 116. The full names of “LkRH53” and “LkCIP7” are “DEAD-box ATP-dependent RNA helicase 53-like protein” and “COP1-interacting protein 7”, respectively.

Comment 2:

Line 96 to Line 107: State the relationship between ARFs and DEAD-box family genes.

Response:

We have done. Please see Line 100-Line 112.

Comment 3:

Line 99: The “*Physcomitrella patens*” and “*Arabidopsis*” in Line 99, the “rh31” in Line 103 and Line 104, the “*Arabidopsis*” in Line 106, the “cop1” in Line 116, should be in italics.

Response:

We have revised them, and checked the font format of all genes and species names in the whole text and made sure that they were italicized.

Comment 4:

Line 102-104: Change the “cold and drought stress” in Line 102 to “cold and drought stresses”. Transgenic Line could be either overexpression Line or loss-of-function line depending on the plasmid you transferred. So the “transgenic line” in Line 104 should change to “the overexpression line” or “the complemental line”.

Response:

We have revised these words, rewritten this paragraph, deleted some sentences and changed the citation. Please see Line 106-107.

Comment 5:

Line 108 to Line117: Elaborate the relationship between ARFs and COP1-

interacting proteins (CIPs) genes.

Response:

The relationship between *ARFs* and *CIPs* in AR formation is not clear, here we found their interactions. We have transferred this paragraph to the Part of Discussion. Please see Line 297-Line 305.

Comment 6:

Line 120: You could not say “LkARF19 interacted with LkRH53 and LkCIP7 to regulate AR formation”, because you did not show any evidence that the interaction of LkRH53 and LkCIP7 with LkARF19 is necessary to its AR formation-function.

Response:

We have modified these sentence and deleted “to regulate AR formation”. Please see Line 115-116.

Comment 7:

Line 162: Change the “LkBBM1 began to be expressed at 8 DAC (day after cutting) and then showed an upward trend” to “the expression of LkBBM1 begins at 8 DAC (day after cutting) and gradually increases”.

Response:

We have done. Please see Line 156-Line 157.

Comment 8:

Line 165 to Line 166: There should be a summary sentence explaining why LkARF7 and LkARF19 were studied.

Response:

We have done. Please see Line 157-159.

Comment 9:

Line 189-191: Change the “The Y187 yeast strain (pHIS2-LkBBM1) was cultured on SD/-Trp-Leu-His (TDO) for screening concentrations of 3-AT and indicated that 75 mM 3-AT was the appropriate concentration” to “The Y187 yeast strain expressing pHIS2-LkBBM1 and pGADT7 empty vector was cultured on SD/-Trp-Leu-His (TDO), and the result showed that 75 mM 3-AT was the appropriate concentration to inhibit the self-activation”.

Change the “We found that two Y187 strains, (pGADT7-LkARF7/pHIS2-LkBBM1 and pGADT7-LkARF19/pHIS2-LkBBM1),” to “the Y187 strain co-expressing pGADT7-LkARF7 and pHIS2-LkBBM1, and the strain co-expressing pGADT7-LkARF19 and pHIS2-LkBBM1”.

Response:

We have changed these sentences. Please see Line 182-186.

Comment 10:

The subcellular of ARF/BBM and the transcription regulation of BBM1 by ARF should be in separated chapters. In the same time, the fig2A and fig2B-C should in

separated figures too. The transcription regulation of BBM1 by ARF could move to the next chapter.

Response:

We have done, and changed Figure 2B and 2C to Figure 3A.

Comment 11:

Line 204-212: This assay is named “dual-luciferase assays”, NOT “dual-luciferase assays and luciferase complementation imaging assays”. CANCEL “in firefly luciferase complementary imaging assays” in Line 210 at the same time.

Response:

We have done. Please see Line 196-197.

Comment 12:

Line 208: Change “a large number of red and white fluorescent signals” to “strong fluorescent signal”.

Response:

We have done. Please see Line 200-201.

Comment 13:

Line 218: Cancel the “In general, the phenotype of LkARF19 OE lines is superior to that of LkARF7 OE lines.”, which is not scientific (how you define “superior”)?

Response:

We have deleted this sentence. Please see Line 210.

Comment 14:

As is shown in fig 4E, the difference of rooting rate between LkARF7-OE, LkARF19-OE lines and Ctrl appeared at 3 DAC, but in Line 162 and figure 1E, you mentioned “LkBBM1 began to be expressed at 8 DAC”, which is far late behind 3 DAC. These data indicating that it is not the BBM1 that should responsible for the rooting rate different caused by ARF7/19. How do you explain it?

Response:

qRT-PCR analysis of *LkBBM1* was carried out with larch material, in which the rooting time of cuttings is ~28 d. *LkARF7*-OE, *LkARF19*-OE lines and Ctrl were poplar material, in which the rooting time were 4-9 d. Due to the difference in materials, the timing of *LkBBM1* expression and poplar rooting is different.

Our current data demonstrated that both *LkARF7/19* and *LkBBM1* promoted poplar AR formation, respectively, and here we found that *LkARF7/19* regulated *LkBBM1*.

Comment 15:

Line 227: The description “The transgenic lines of OE LkARF7 and LkARF19 were larger than Ctrl” is not scientific and precise. How you define LARGER of a seedling? The leaf? Root length? Width of stem? Or the height?

Response:

It was height, we have revised these sentences. Please Line 216-Line 221.

Comment 16:

Line 240: Change “Figure 9A” to “Figure 6A”.

Response:

We have done. Please Line 225.

Comment 17:

Line 255: “Because the interaction between LkARF19 and LkRH53 is stronger (than CIP7)”. but we could not conclude it from the Y2H data and BIFC data.

Response:

We have deleted this sentence.

Comment 18:

Line 259: “LkCIP7 is involved in regulating AR formation.” in fact, you do not show sufficient evidence to get this conclusion. The protein interaction data and the expression pattern of CIP7 is not sufficient.

Response:

We have deleted this sentence.

Comment 19:

The paper contains repeated points in the different sections: Results-Methods. Please modify it appropriately.

Response:

We have deleted some sentences, removed the repeated points and simplified some statements. Please see Line 138-139, Line 148-150, Line 175-176, and Line 181.

Comment 20:

Line 378: At what time were the roots, leaves, stems, and stem tips collected.

Response:

They were collected from two-month-old seedlings, and we have added this information in Part of Materials and methods. Please see Line 319.

Comment 21:

Line 382: What is the method of poplar stable transformation? Describe the method, or provide relevant references.

Response:

We have added the poplar stable transformation method in Part of Materials and methods. Please see Line 324-Line 325.

Comment 22:

Line 438: What are the culture conditions of tobacco?

Response:

We have added the tobacco culture conditions in Part of Materials and methods. Please see Line 325-Line 328.

Comment 23:

Line 455 to Line 457: This sentence is not clearly described. It is suggested to describe the transient expression and ChIP methods separately.

Response:

We have divided the content of this paragraph into two parts: Transient transformation assay of larch and Chromatin immunoprecipitation (ChIP) assay. Please see Line 380-Line 388 and Line 397-Line 404.

Comment 24:

In Figure 1, what software is used to make the heat map, what are the parameters?

Response:

We have added software instructions and the parameters in Figure1 legend and the Part of Materials and methods. Please see Line 449-Line 451.

Reviewer #2

Reviewer's summary-comment:

This manuscript describes ARF transcription factors as important auxin-induced regulators upstream of BBM in the regulation of adventitious root formation in *Larix kaempferi*, a woody species of economical interest. The authors state that auxin is important for the formation of adventitious roots and they have earlier identified BABYBOOM as an important regulator of AR formation in Larix.

The data in the manuscript add to the model that certain ARFs (7-19) interact directly with BBM to regulate its transcription levels and are hence involved in AR formation. It furthermore places 2 other unknown transcription regulators in the AR pathway.

The experiments are sufficiently well executed and data are correctly represented. A reasoning for transforming poplar and not larch is lacking so the importance of the phenotypes observed in poplar is unclear.

The article is well written and does not contain a lot of mistakes. The authors should however, check consequently if genes and mutants are written in italic and proteins not. Throughout the manuscript there are mistakes in this formatting. My major problem is the lack of novelty in the conclusion, since ARFs are already long-known to be involved in adventitious root formation and in their role as direct transcription regulators upon auxin perception it is logic that they would act upon different other transcription factors that regulate cell divisions and identity necessary for AR formation (such as BBM).

I have included some suggestions, questions and comments below. Certain comments are essential to be answered for the manuscript to be published. Moreover, I think the authors would benefit from reconsidering the submission of this manuscript to a more specialized plant journal where the visibility will be higher and hence the impact for the scientific community will increase as well.

Abstract

Comment 1:

I think the termination cutting seedling is confusing.

Response:

We have changed “cutting seedling production” to “Cuttage propagation”. Please see Line 31.

Comment 2:

l38 remove that.

Response:

We have deleted “that”. Please see Line 38.

Comment 3:

l40 expression is not the correct word. The interaction with ARFs promotes transcription of a gene.

Response:

We have changed “expression” to “transcription”. Please see Line 40.

Comment 4:

Line 45: “... novel regulatory mechanism underlying the control of AR formation by auxin IN LARIX KAEMPFERI”. To me the statement, downstream of auxin signaling is not clear enough - are known auxin receptors studied in Larix? ARFs are known to act downstream of auxin perception but they are a major part of auxin signaling so I would never name this downstream of auxin signaling.

Response:

We have added “in *Larix kaempferi*” in this sentence. Please see Line 44-Line 46. Auxin receptors and its signaling pathway in *Larix* were not studied.

Comment 5:

keywords: all genes should be italic (I agree that above where you mention interaction it is okay that they are not italic since there you speak on interacting proteins).

Response:

We have changed these and checked the font format of all gene names in the whole text and made sure that they were italicized.

Introduction

Comment 6:

Line 56-57: it is unclear what is meant by cell recombination and formation.

Response:

We have modified these sentences. Please see Line 55-Line 57.

Comment 7:

Line 57: it is better not to start a sentence with and.

Response:

We have deleted it. Please see Line 57.

Comment 8:

Line 61: auxin response FACTORS (never singular unless a specific one is indicated in the sentence).

Response:

We have modified this sentence. Please see Line 61-Line 62.

Comment 9:

Line 64: if you write receiving here, it is as if ARFs are perceiving the auxin signal. That is not true, they act downstream of auxin receptors (TIR1/AFBs).

Response:

We have deleted “after receiving the auxin signal”. Please see Line 64.

Comment 10:

Line 75: “In addition, IN ARABIDOPSIS THALIANA”, I think the species should be added since it is confusing when model and woody species are mixed.

Response:

We have added “in *Arabidopsis*” in this sentence. Please see Line 72.

Comment 11:

Line 79: it is not true that the underlying mechanisms are not known. Probably for woody species this statement is more correct, but for models such as *Arabidopsis thaliana*, different approaches have identified important downstream actors (for example: Lee et al, 2019 BMC Plant Biology; Li 2021, Front Plant Sci; Yin et al, 2020 Plant Phys).

I do think that this statement should be removed or specified. Also, the lack of references to literature in lateral or adventitious root formation and involved ARFs, indicates that the authors might need to do some more background research in the field to well establish their novelty and the importance of their research compared to what is done and known from other species.

Response:

We have changed this statement and added “in woody plants”.

We have also added more studies on ARFs involvement in adventitious root formation from other model plants. Please see Line 75-Line 82.

Comment 12:

Line 81 mention clearly that BBM is a transcription factor, which regulates totipotency in plant cells and hence has a known role in promoting cell proliferation and morphogenesis during embryogenesis.

Response:

We have modified it. Please see Line 84-86.

Comment 13:

Line 84: mention clearly that BBM is found to act upstream of the other known embryogenesis transcription factors.

Response:

We have modified it. Please see Line 89-90.

Comment 14:

Line 88: change well to a scientifically more correct description of the response to auxin.

Response:

We have changed “well” to “strong”. Please see Line 92.

Comment 15:

Line 93: instead of the conclusion stated:

Since ARFs are known regulators of auxin response and BBM acts downstream of

auxin induction during AR formation, we investigate the potential relationship between BBM and ARFs.

Response:

We have modified it. Please see Line 97-Line 99.

Comment 16:

Line 96-117: the link to RH and COP1 is not sufficiently explained and not sufficiently smooth or clear.

Maybe add something among these lines: In order to investigate potential processes involved in AR formation downstream of BBM, we investigated RH and COP1.

Response:

We have modified the sentences related to RH. Information about COP1 was transferred to the Part of Discussion. Please see Line 100-Line 112 and Line 297-Line 305.

Results

Comment 17:

Line 125: ... which genes regulate LkBBM1... (i hope the authors were aiming to identify multiple genes and not just one).

Response:

We have modified this sentence. Please see Line 120.

Comment 18:

Line 130: ... suggesting that ARF genes COULD be involved in the regulation OF TRANSCRIPTION of LkBBM1.

Response:

We have done. Please see Line 125.

Comment 19:

Line 143: you mention members of the gene family, but construct a tree using protein sequences  delete gene from your sentence.

Response:

We have done. Please see Line 137-Line 138.

Comment 20:

Line 188-195: which controls were used for this Y1H setup?

Response:

The combination of p53HIS2 and pGAD53T was used as positive control. The combination of pHIS2-LkBBM1 promoter and pGADT7 was used as negative control. These have been illustrated in the Part of Materials and methods (in Line 393-Line 395) and Figure 3.

Comment 21:

Line 204: remove the.

Response:

We have done. Please see Line 196.

Comment 22:

Line 246: for the BIFC assay a negative control ARF would be of interest.

Response:

Here, two groups of negative controls were used: LkARF19-cYFP + nYFP and cYFP + LkRH53-nYFP. Please see the instructions in Figure 6 legend.

Comment 23:

Line 252: how significant is an induction of 1.84x?

Response:

Based on the T-test, *P* value was less than 0.05 and the difference was significant. This analysis could be found in instructions of Figure 6. We have modified this sentence. Please see Line 239-Line 240.

Comment 24:

Line 278: the induction of auxin on BBM expression does not indicate a direct downstream effect. To conclude this, one should show that either.

- auxin biosynthetic mutants express less BBM.
- in an auxin receptor-mutant (*tir1/afb*) there is no auxin effect on BBM expression.

Response:

Based on your comment and suggestion from Reviewer 3, this part seems to have diverged from the main point, so we deleted the first Part of Discussion.

Comment 25:

Line 284: ARFs, plant specific transcription factors, regulate....

Response:

We have done. Please see Line 248.

Comment 26:

Line 302-304: this is indeed the explanation of auxin signaling and I would mention this much higher up in your manuscript's introduction or when you introduce ARFs as potential expression regulators.

Response:

We have removed this sentence and made some modifications in Part of Introduction.

Comment 27:

Line 305: 2h is not rapidly.

Response:

We have modified and deleted “rapidly” in this sentence. Please see Line 265-Line

266.

Comment 28:

Line 308: expression IS affected.

Response:

We have done. Please see Line 263.

Comment 29:

Line 309: These DATA suggest that...

Response:

We have done. Please see Line 267-Line 268.

Comment 30:

Line 312-313: Previous studies have shown that auxin signaling is required in the induction stages of LR and AR.

Response:

We have deleted “a strong” in this sentence. Please see Line 270-Line 271.

Comment 31:

Line 313-315: Our data in poplar are consistent with this, since overexpression of LKARF7 and LKARF19 significantly increase the AR numbers.

Response:

We have replaced “These are consistent with results of the earlier formation and greater number of ARs in LkARF7- and LkARF19-OE lines in 84K poplar” by “Our data in poplar are consistent with this, since overexpression of *LkARF7* and *LkARF19* significantly increase the AR numbers”. Please see Line 274-Line 276.

Comment 32:

Line 320: remove momentous and replace by an important.

Based on this statement, did the authors observe shoot growth phenotypes in the overexpression poplar lines to substantiate this statement?

Response:

We have done. Please see Line 276. We indeed observed shoot growth phenotypes in the overexpression poplar lines to substantiate this statement by investigating plant height, ground diameter, and internode number.

Comment 33:

Line 329: see earlier, I find this an overstatement since downstream molecular factors are known.

Response:

We have deleted this paragraph, and made some modifications in Part of Introduction.

Comment 34:

Line 332-334: expression pattern points more commonly to cellular expression and not to timely expression. Please specify because your data point to a timely relationship in expression of both factors.

Response:

We point to timely expression. We have modified this sentence. Please see Line 282-Line 283.

Comment 35:

Line 337-343: as you state here, a lot of downstream genes in the BBM signaling are known and loads of them have an important action in root development. Hence the statement of a lack of molecular mechanisms is untrue and should be related to this or mentioned specifically for larch.

Response:

We have modified these sentences. Please see Line 285-Line 286.

Comment 36:

Line 344: It remains unclear why these targets were chosen and not more other logic root-developmental genes as SHR or SCR that were mentioned before.

Response:

By Y2H assays, only these the two genes were screened, and other logic root-developmental genes as SHR or SCR were not identified. This also encourages us to screen more interacting proteins in future. In addition, the molecular mechanism of LkRH53 and LkCIP7 in AR formation in larch is also needed to study.

Comment 37:

Figure 6: Figure outline is not consistent with the other graphs in the manuscript, they have not received a border (check and adjust everywhere accordingly). BIFC interaction is not sufficiently clear due to the lack of signal in the YFP channel.

Response:

We have adjusted Figure 6B, and now the YFP fluorescence signals can be clearly seen, please check.

Reviewer #3

Reviewer's summary-comment:

The manuscript entitled "Larch transcription factors LkARF7 and LkARF19 promote adventitious root formation through positive regulation of LkBBM1" by Tao *et al.* describes the characterization of the auxin response factor (ARF) family of *Larix kaempferi* × *L. olgensis* and identifies LkBBM1 as a target gene of two members of this family. LkBBM1 is a transcription factor of the AP2/ERF family, homologous to the BBM gene of *Arabidopsis* with recognized function in somatic embryogenesis. The participation of LkBBM1 in larch adventitious rooting has been demonstrated in a previous work. In addition, the function of both ARFs in adventitious rooting is studied, using poplar as an experimental system. Their interaction with two other genes is also described.

The experimental work is correct, and the methodology is sound. However, certain weaknesses have been detected that should be reviewed before proceeding to publication.

Comment 1:

DAPI staining is missing in Figure 2A. In addition, the results described in the text corresponding to Figures 2B and 2C are not consistent with those represented in the Figures.

Response:

We have done the subcellular localization experiment again, and added DAPI fluorescence signal. Please see Figure 3.

The description of the result in Figures 2B and 2C (Now becomes Figure 3A) is written by mistake; we have corrected them.

Comment 2:

It is mandatory to include an additional empty vector control in the poplar gene transformation assays (Figures 4, 5, 6 and S5). Transformations with empty vectors can increase rooting ability in certain species.

Response:

Based on this suggestion, transformations with empty vectors were performed, and the data was analyzed again. The results showed that almost no difference was found in the number of AR between WT and lines with empty vectors.

Comment 3:

The increase in biomass, measured as fresh weight or dry weight of the aerial part of the plant and of the root system as a whole, may be a consequence of the increase in the number and length of roots, which could indirectly favor stem growth.

No analyses on specific biomass indicators are described that would allow us to conclude on the effect of these genes on biomass beyond the increase in the number and length of roots.

Response:

In this manuscript, the fresh or dry weight of the aerial part of the plant were

analyzed. Based on the results, we concluded that biomass increased after overexpressing these two *LkARFs*.

Notably, our results quite agree with your viewpoint, because in the transgenic lines the number and length of AR indeed increased, which might be a reason of increase of biomass of the aerial part of the plant. Since, the theme of this work is that *LkARF7* and *LkARF19* promote AR formation, we have placed Figure 5 in the supplement files (Figure S6).

Comment 4:

The image quality in Figure 6B should be improved. For example, the YFP is difficult to see.

Response:

We have adjusted Figure 6. The YFP fluorescence signals can be clearly seen in Figure 6B.

Comment 5:

Some sections of the discussion seem to have diverged from the main point. Please, streamline the discussion by removing items that are not directly related to the data and consider removing this content from the discussion section that is more appropriate to be in the introduction section.

Response:

We have revised the discussion based on your suggestion. We have deleted the paragraph titled “*LkBBM1 acts as a marker gene for AR primordia and regulates AR formation*” and the discussion about the target genes of *BBM*.

We also deleted some information about the role of *ARFs* and the effect of auxin concentration on *ARFs* expression from the Part of Discussion and put them in the Part of Introduction.

Reviewers' comments:

Reviewer #3 (Remarks to the Author):

The manuscript entitled "Larch transcription factors LkARF7 and LkARF19 promote adventitious root formation by positively regulating of LkBBM1" by Tao et al. has been revised according to my comments.

However, it still needs modifications before proceeding to publication:

1. In the legend of Figure 2, Figures A and B are mentioned. However, they are neither mentioned in the text nor indicated in the figure. The sections can be omitted since the names of the genes are indicated in the figure.
2. The text of the figure legend S5 should be revised. Should it indicate "by amplification of genomic DNA" instead of "by genome DNA"?
3. Figures S6B and S6C are interchanged in the legend of figure. It should be corrected to avoid confusion.
4. Should Figure S4 be mentioned instead of Figure S2 on line 266?
5. Should another figure be mentioned instead of Figure 1 on line 281?
6. The protocol for transformation and regeneration of the overexpression lines should be detailed in the corresponding material and methods section, especially the time required for each step. The included reference does not detail it.
7. The manuscript should be read carefully to correct verbal forms, other grammatical and spelling errors.

Reviewer #4 (Remarks to the Author):

Since I have specifically asked to focus on the comments of reviewer #2, I strongly agree with this reviewer's position that there is only limited conceptual novelty in the report. As this reviewer pointed out, this I because it was already known from previous studies that auxin and larch BBM promote AR formation and because there is a huge body of literature demonstrating that ARF transcription factors mediate the transcriptional output of auxin. The authors addressed this concern of reviewer #2 only in the sense that they toned down the novelty claims in the text. I also agree that this report may fit better in a plant specific journal.

This reviewer's comment on why gain-of-function experiments were done in poplar rather than in larch was left unanswered. Transformation of larch is well established, and without convincing explanation why a heterologous system was used to study the phenotypical readout of these genes, this experimental setup appears a weakness.

Although I have not done an in-depth-review of the whole paper, two more points drew my attention.

First, for protein-protein and for protein-DNA experiments, the authors entirely rely on overexpression strategies. Although it is clear that increased concentrations can produce wrong interaction results, the authors seem to take the obtained data as proof. I think it is necessary to address this issue experimentally before conclusions can be drawn.

Second, in the final section new players are introduced as interactors of these ARFs. Yet the functional relevance of this interaction is not addressed and therefore, this part appears immature for publication.

We sincerely express our heartfelt thanks to the Reviewers for their technical comments, and extensive and thoughtful comments. We have thoroughly revised the manuscript according to the Reviewers' comments and their valuable suggestions and addressed all the concerns raised.

Reviewer #3

Reviewer's summary-comment:

The manuscript entitled "Larch transcription factors LkARF7 and LkARF19 promote adventitious root formation by positively regulating of LkBBM1" by Tao et al. has been revised according to my comments.

However, it still needs modifications before proceeding to publication:

Comment 1:

In the legend of Figure 2, Figures A and B are mentioned. However, they are neither mentioned in the text nor indicated in the figure. The sections can be omitted since the names of the genes are indicated in the figure.

Response:

We have deleted these sections in the legend of Figure 2.

Comment 2:

The text of the figure legend S5 should be revised. Should it indicate "by amplification of genomic DNA" instead of "by genome DNA"?

Response:

We have done, we changed the order of the supplement figures, so that Figure S5 became Figure S3.

Comment 3:

Figures S6B and S6C are interchanged in the legend of figure. It should be corrected to avoid confusion.

Response:

We have done, please see the legend of Figure S4.

Comment 4:

Should Figure S4 be mentioned instead of Figure S2 on line 266?

Response:

We have replaced "Figure S2" to "Figure S5". Please see Line 310.

Comment 5:

Should another figure be mentioned instead of Figure 1 on line 281?

Response:

We have replaced "Figure 1" to "Figure S6". Please see Line 317-Line 318.

Comment 6:

The protocol for transformation and regeneration of the overexpression lines should be detailed in the corresponding material and methods section, especially the time required for each step. The included reference does not detail it.

Response:

We have added detailed steps for poplar transformation. Please see Line 348-Line 356.

Comment 7:

The manuscript should be read carefully to correct verbal forms, other grammatical and spelling errors.

Response:

Thanks for the comments. We have revised the language of the whole manuscript.

Reviewer #4

Reviewer's summary-comment:

Since I have specifically asked to focus on the comments of reviewer #2, I strongly agree with this reviewer's position that there is only limited conceptual novelty in the report. As this reviewer pointed out, this is because it was already known from previous studies that auxin and larch BBM promote AR formation and because there is a huge body of literature demonstrating that ARF transcription factors mediate the transcriptional output of auxin. The authors addressed this concern of reviewer #2 only in the sense that they toned down the novelty claims in the text. I also agree that this report may fit better in a plant specific journal.

Comment 1:

This reviewer's comment on why gain-of-function experiments were done in poplar rather than in larch was left unanswered. Transformation of larch is well established, and without convincing explanation why a heterologous system was used to study the phenotypical readout of these genes, this experimental setup appears a weakness.

Response:

Gene overexpression and silencing can be achieved in larch somatic embryos or protoplasts. However, it is challenging to obtain seedlings after transforming larch somatic embryos. We have added some instructions to the text. Please see Line 298-Line 301.

Comment 2:

Although I have not done an in-depth-review of the whole paper, two more points drew my attention.

First, for protein-protein and for protein-DNA experiments, the authors entirely rely on overexpression strategies. Although it is clear that increased concentrations can

produce wrong interaction results, the authors seem to take the obtained data as proof. I think it is necessary to address this issue experimentally before conclusions can be drawn.

Response:

We absolutely agree with your comments. For protein-protein experiment, we conducted Yeast two-hybrid and Bimolecular fluorescence complementation (BiFC) assays. For protein-DNA experiment, we conducted Yeast one-hybrid, Chromatin immunoprecipitation (ChIP) and Dual luciferase assays. We think these experiments can lead to these conclusions.

Comment 3:

Second, in the final section new players are introduced as interactors of these ARFs. Yet the functional relevance of this interaction is not addressed and therefore, this part appears immature for publication.

Response:

Based on the comment and suggestion, we have removed the relevant content of interaction between LkARF19 and LkCIP7.